# DUALRPO: ALL-IN-ONE VISUAL RL WITH INTERNAL AND EXTERNAL REWARDS

## ABSTRACT

Reinforcement learning (RL) has primarily advanced the reasoning capabilities of Vision-Language Models (VLMs) in multi-modal scenarios, and some recent works have explored using RL to enhance their perception abilities. However, developing a unified RL framework to handle both task types simultaneously confronts a critical bottleneck: task interference driven by heterogeneous tasks. We observe that the interference typically manifests as training instability and ambiguous responses, ultimately constraining the effectiveness of unified multi-task training. To address this challenge, we propose **DualRPO** (Dual Rewards Policy Optimization), a novel RL paradigm that synergistically integrates internal self-certainty rewards and external verifiable rewards. DualRPO embeds self-certainty, defined as the average KL divergence between the model's output distribution and a uniform distribution, in reward shaping: *it amplifies external rewards for correct yet underconfident outputs and penalizes external rewards for incorrect but over-confident ones*, guiding the model to generate accurate and confidence-calibrated responses. Extensive experiments validate the efficacy of DualRPO. We evaluate across 8 heterogeneous tasks (5 perception: chart analysis, detection, grounding, counting, OCR; 3 reasoning: math, puzzle, science). DualRPO delivers a large performance improvement across all tasks, with training instability amplitude reduced by. These results highlight that the proposed DualRPO enables unified scaling of multi-modal models to diverse perceptual and cognitive tasks.

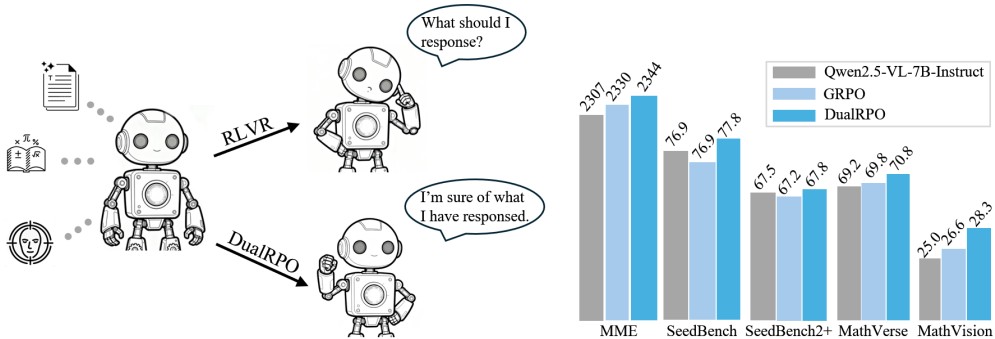

Figure 1: Overview of DualRPO and Performance. Left: Illustration of DualRPO for unified training, which force the model to generate confident responses. Right: Performance comparison of Qwen2.5-VL-7B-instruct, GRPO, and DualRPO. Both GRPO and DualRPO trained on the same datasets. DualRPO elevates the base model to a higher level compared with GRPO.

## 1 INTRODUCTION

The field of reinforcement learning (RL) has recently witnessed remarkable progress in enhancing the capabilities of large language models (LLMs) and vision-language models (VLMs). A pivotal advancement in this space is Reinforcement Learning with Verifiable Rewards (RLVR), which moves beyond learned reward models by leveraging automatically verifiable signals, such as exact

answer matching in mathematical reasoning—to drive significant improvements in reasoning performance (Chen et al. (2025b); Shen et al. (2025a); Chen et al. (2025a); Shen et al. (2025b); Meng et al. (2025); Huang et al. (2025); Yang et al. (2025); Liu et al. (2025a); Wang et al. (2025c;b); Xia et al. (2025); Xiao et al. (2025); Zhu et al. (2025); Wan et al. (2025); Yao et al. (2025)).

Recent works in RLVR focus on reasoning tasks (e.g., mathematics) but has limited use in multi-modal perception, e.g., Shen et al. (2025b); Liu et al. (2025d) use IoU as external rewards for tasks like object detection. Yet a critical gap persists: few unify these heterogeneous perceptual and cognitive tasks in one RL pipeline to enable VLMs to excel at both. Prior work like V-Triune (Ma et al. (2025)) advances this via standardized formatting and verifier-level rewards, achieving stable optimization across some perception-reasoning tasks. Yet task interference remains a bottleneck in unified VLM training: competing tasks cause instability (e.g., fluctuating losses), low confidence, and poor generalization—undermining versatile multi-modal model development.

To address this challenge, we propose **DualRPO** (Dual Rewards Policy Optimization), a novel RL paradigm that synergistically integrates internal self-certainty rewards and external verifiable rewards for VLMs. Our motivation stems from a key empirical observation: *VLMs trained on heterogeneous multi-task datasets consistently exhibit lower response confidence compared to models trained on single-domain data.* This confidence degradation is directly linked to task interference, which amplifies entropy loss during optimization and disrupts the model's ability to form stable task-specific representations.

DualRPO mitigates this interference via a dual-modulation design centered on self-certainty, defined as the average KL divergence between the model's output distribution and a uniform distribution (Fang et al. (2024); Kang et al. (2025); Zhao et al. (2025)). First, it uses paired modulation coefficients for reward shaping: one amplifies external rewards for correct low-confidence outputs, while the other penalizes them for incorrect overconfident ones, ensuring the model produces both correct and confidence-calibrated results. Second, it leverages self-certainty to dynamically adjust the KL penalty between **0.5** and **1**: the penalty is relaxed for high output confidence (to encourage exploration of diverse solutions) and strengthened for low confidence (to enforce optimization stability and avoid divergence). By linking KL weight directly to self-certainty, this mechanism balances exploration and regularization, effectively boosting the robustness of unified multi-task training.

To provide a clear overview of DualRPO's design and performance, Fig. 1 presents that the DualRPO forces the model to generate confident responses in the unified training.

Extensive experiments validate the effectiveness of our method. Using Qwen2.5-VL-7B-Instruct (Bai et al. (2025)) as the base model, we apply unified reinforcement training across two main task categories: reasoning and perception, encompassing eight diverse tasks in total (math, puzzle, science, chart, detection, grounding, counting, and OCR). As shown in Fig. 1 (right panel), DualRPO achieves consistent performance gains across all 6 benchmarks. Notably, these gains outperform single-reward baselines by a significant margin, confirming that fusing internal and external rewards is critical for addressing task interference. The results further demonstrate that DualRPO substantially improves both perception and reasoning performance, highlighting the promise of confidence-guided reward modulation as a core principle for advancing unified reinforcement learning in VLMs.

Our contributions can be summarized as follows:

- To our knowledge, we are the first to systematically integrate internal self-certainty signals with external verifiers for vision–language model RL. We propose **DualRPO**, which mitigates task interference via self-certainty modulation of both reward function and KL loss, enabling stable multi-task learning.

- We conduct rigorous scenario-aware analysis of internal signals' utility (unified training, perception-only, reasoning-only settings), providing guidelines for when internal signals can serve as the sole reward in multi-modal RL.

- Extensive experiments on 8 heterogeneous sub-tasks show **DualRPO** stabilizes unified RL and advances the GRPO baseline on 8 benchmarks, validating the value of fusing internal/external rewards for multi-modal model scaling.

## 2 RELATED WORK

**Reinforcement Learning with Verifiable Rewards (RLVR).** Reinforcement Learning with Verifiable Rewards (RLVR) has emerged as an alternative to reinforcement learning from human feedback (RLHF), aiming to eliminate the reliance on costly reward models. Instead of learning preferences from annotated data, RLVR leverages automatically verifiable signals such as exact answer matching in mathematics, program execution correctness in code, or structured schema validation. This paradigm has proven particularly effective in reasoning-intensive domains(Guo et al. (2025); Chen et al. (2025b); Shen et al. (2025a;b); Meng et al. (2025); Huang et al. (2025); Yang et al. (2025); Liu et al. (2025a); Wang et al. (2025c;b); Xia et al. (2025); Xiao et al. (2025); Zhu et al. (2025); Wan et al. (2025); Yao et al. (2025)), where objective correctness can be automatically determined, thus avoiding the biases and inconsistencies of human supervision. However, RLVR is less straightforward to apply in perception and open-ended tasks, where gold-standard verification is unavailable.

**Internal Reward for Large Language Models.** Learning from internal rewards has emerged as a powerful paradigm for model self-improvement. Chen et al. (2024) and Yuan et al. (2024) employ the model itself to generate feedback for iterative optimization. Leveraging procedural generalization, Poesia et al. (2024) demonstrate self-improvement within formal domains using only axioms as supervision. Cheng et al. (2024) propose SPAG, where LLMs act as both defenders and attackers in an adversarial setup to refine their capabilities. More recently, Wang et al. (2025b); Zhao et al. (2025) report remarkable gains by adopting entropy loss or self-certainty as internal rewards, serving as the sole optimization signal to unlock hidden model capabilities without relying on gold solutions or test cases. Different from these works, we observe that relying solely on internal rewards often leads to model collapse in unified reinforcement training, particularly in tasks requiring structured output formats. To achieve robust optimization, we instead integrate internal rewards as guidance signals for RLVR, mitigating task interference by enhancing the model's confidence in answering specific categories of problems.

**Unified Training in Vision-Language Models.** Prior work has applied reinforcement learning to isolated domains such as mathematical reasoning or detection. Huang et al. (2025); Peng et al. (2025); Yang et al. (2025); Liu et al. (2025b) are among the first to explore the feasibility of RLVR in VLMs. Liu et al. (2025c) establish a unified framework spanning three tasks—detection, segmentation, and counting—but its applicability to broader domains is hindered by inconsistent rule-based rewards. To address this limitation, Ma et al. (2025) propose V-triune, which combines sample-level data formatting with verifier-level reward computation, enabling large-scale training across diverse tasks. Nonetheless, task interference remains a core challenge in unified training due to reward misalignment. To tackle this, we leverage self-certainty as an internal signal to modulate external rewards, providing a consistent supervisory signal that mitigates interference across heterogeneous reward rules.

## 3 METHOD

In this section, we present our approach to unified reinforcement training on vision language models through internal guided external reward. We begin by reviewing existing RL-based fine-tuning paradigms and their limitations in unified reinforcement training, which motivate our exploration of Reinforcement Learning from Internal Feedback (RLIF). We then introduce DualRPO, leveraging self-certainty as an internal reward signal to guide the external reward, and detail its implementation through policy optimization.

### 3.1 EXTERNAL AND INTERNAL REWARD

We begin by introducing Reinforcement Learning with Verifiable Rewards (RLVR). Unlike Reinforcement Learning from Human Feedback (RLHF), which relies on a learned reward model, RLVR replaces it with automatically verifiable signals. It avoids the burden of rewarded model, and also eliminates the pitfalls introduced by it. where $q$ is an input query, $o$ is the generated output, $\pi_{\text{ref}}$ is an initial reference policy, and $\beta$ is a coefficient controlling the KL divergence to prevent excessive deviation from $\pi_{\text{ref}}$. Online RL algorithms generate samples from $\pi_\theta$, evaluate them using the verifiable reward function $v(q, o)$, and update $\pi_\theta$ to maximize this objective. Due to the lack of gold-standard answers in other fields, RLVR is often applied on math or code tasks.

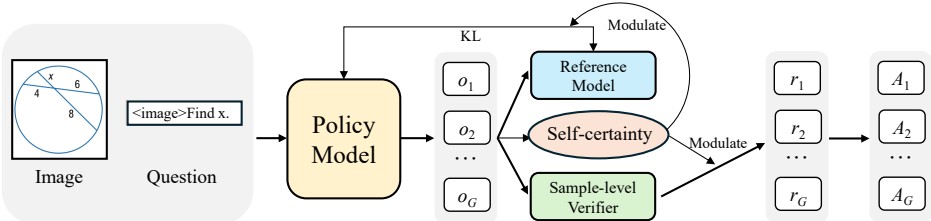

Figure 2: Illustration of DualRPO objective, which extends GRPO by adding the self-certainty as the modulation factor for stable and general training.

$$\max_{\pi_\theta} \ \mathbb{E}_{o \sim \pi_\theta(q)} \Big[ v(q, o) \ - \ \beta \, \mathrm{KL} \big( \pi_\theta(o \mid q) \,\|\, \pi_{\mathrm{ref}}(o \mid q) \big) \Big], \tag{1}$$

Reinforcement learning from internal feedback (RLIF) is another way to reward the samples. It does not need any costly human annotation and domain-specific supervision since it stems from the model's self-certainty. Under the RLIF paradigm, the optimization objective becomes:

$$\max_{\pi_\theta} \ \mathbb{E}_{o \sim \pi_\theta(q)} \Big[ u(q, o) - \beta \, \mathrm{KL} \big( \pi_\theta(o \mid q) \,\|\, \pi_{\mathrm{ref}}(o \mid q) \big) \Big], \tag{2}$$

where $u(q, o)$ represents an internal signal derived from the model's internal state or computation, rather than external verification. The key challenge lies in identifying internal signals that correlate with output quality and can effectively guide learning. Zhao et al. (2025) use the self-certainty as the internal signal, defined as the average KL divergence between a uniform distribution $U$ over the vocabulary $\mathcal{V}$ and the model's next-token distribution:

$$\textbf{Self-certainty}(o \mid q) := \frac{1}{|o|} \sum_{i=1}^{|o|} \mathrm{KL}(U \,\|\, p_{\pi_\theta}(\cdot \mid q, o_{<i})) = -\frac{1}{|o| \cdot |\mathcal{V}|} \sum_{i=1}^{|o|} \sum_{j=1}^{|\mathcal{V}|} \log \big( |\mathcal{V}| \cdot p_{\pi_\theta}(j \mid q, o_{<i}) \big)$$
$$\tag{3}$$

where $o_{<i}$ are the previously generated tokens and $p(j|q, o_{<i})$ is the model's predicted probability for token $j$ at step $i$. Higher self-certainty values indicate greater confidence. Self-certainty, being related to a KL divergence where the model's prediction is the second argument $\mathrm{KL}(U \,\|\, p_{\pi_\theta})$, is mode-seeking. Optimizing for self-certainty thus encourages the model to generate responses that it deems more convincing.

## 3.2 INTERNAL SIGNAL GUIDED EXTERNAL REWARD

Relying solely on external rewards leads to unstable training in unified reinforcement learning. Due to heterogeneous data sources and misaligned reward functions, optimization often becomes biased toward easier objectives while neglecting more challenging examples. This can be observed in practice: samples from perception datasets often receive higher and stable rewards than those from reasoning datasets.

To address this issue, we introduce internal signals to modulate external rewards. The key idea is to combine the reliability of external verifiable signals with the stability of internal signals. In particular, we leverage self-certainty as a confidence-based modulation factor. Correct but unconvincing responses receive amplified rewards, guiding the model toward optimization for difficult samples. Conversely, incorrect but overconfident predictions are penalized, discouraging misplaced certainty. If all responses in one group are incorrect, the modulation naturally degrades to a purely internal signal. When used as the sole reward source, internal reward requires no external supervision but unifies diverse tasks through the model's own confidence. It manifests the same scale and reward standard, maintaining training stability.

Formally, let $v(q, o)$ denote the verifiable external reward and $u(q, o)$ denote the self-certainty. We define a certainty-modulated reward:

$$r(q, o) \;=\; v(q, o) \cdot \big(1 + \mu\, u(q, o)\big) \;-\; \lambda\, (1 - v(q, o))\, u(q, o), \tag{4}$$

$$\hat{A}_{i,t} = \frac{r_i - \mathrm{mean}(\{r_1, r_2, \cdots, r_G\})}{\mathrm{std}(\{r_1, r_2, \cdots, r_G\})}. \tag{5}$$

where $\mu$ and $\lambda$ are coefficients that control the amplification of correct but uncertain outputs and the penalization of incorrect yet confident ones, respectively.

Besides, we use the self-certainty to modulte the KL loss. The modulation fomula is as following:

$$\beta(c) = \beta_0 \Big(\beta_{\min} + (1 - \beta_{\min})(1 - c)\Big), \tag{6}$$

where $c$ is the normalized self-certainty score of the model's output. $\beta_0$ is the base scaling factor for the KL term. $\beta_{\min}$ is the minimum KL weight that prevents the coefficient from vanishing. When the model is more certain (higher $c$), the KL penalty $\beta(c)$ decreases, allowing the model to explore more freely. When the model is less certain (lower $c$), the KL penalty $\beta(c)$ increases, constraining the policy update and stabilizing training. This modulation ties the strength of KL regularization directly to the model's confidence, helping balance exploration and stability in unified training.

Compared with pure RLVR, this formulation introduces a lightweight, confidence-aware modulation that stabilizes unified training across heterogeneous datasets. By integrating self-certainty into reward shaping, the model is encouraged not only to maximize verifiable correctness but also to calibrate its confidence, thereby mitigating task interference and preserving knowledge across tasks. For each query $q \sim P(Q)$, GRPO samples a group of G outputs $o1, ..., o_G$ using a behavior policy $\pi_{\theta_{\mathrm{old}}}$. The target policy $\pi_\theta$ is then optimized by maximizing:

$$\mathcal{J}_{\mathrm{GRPO}}(\theta) = \mathbb{E}_{q \sim P(Q), \{o_i\}_{i=1}^{G} \sim \pi_{\theta_{\mathrm{old}}}(O|q)} \tag{7}$$

$$\left[ \frac{1}{G} \sum_{i=1}^{G} \frac{1}{|o_i|} \sum_{t=1}^{|o_i|} \Big( \min\Big[ c_{i,t}(\theta)\hat{A}_{i,t},\ \mathrm{clip}\big(c_{i,t}(\theta), 1 - \epsilon, 1 + \epsilon\big)\hat{A}_{i,t} \Big] - \beta(c)\, \mathbb{D}_{\mathrm{KL}}(\pi_\theta \| \pi_{\mathrm{ref}}) \Big) \right].$$

$$c_{i,t}(\theta) = \frac{\pi_\theta(o_{i,t} \mid q, o_{i,<t})}{\pi_{\theta_{\mathrm{old}}}(o_{i,t} \mid q, o_{i,<t})}, \mathbb{D}_{\mathrm{KL}}(\pi_\theta \| \pi_{\mathrm{ref}}) = \frac{\pi_{\mathrm{ref}}(o_{i,t} \mid q, o_{i,<t})}{\pi_\theta(o_{i,t} \mid q, o_{i,<t})} - \log \frac{\pi_{\mathrm{ref}}(o_{i,t} \mid q, o_{i,<t})}{\pi_\theta(o_{i,t} \mid q, o_{i,<t})} - 1.$$

## 4 EXPERIMENTS

**Base model and dataset.** We adopt Qwen2.5-VL-7B-instruct(Bai et al. (2025)) as out base model, which is pretrained by approximately 4 trillion tokens. For training the unified model, we adopt the datasets curated from V-triuneMa et al. (2025), which contains eight different tasks: math, puzzle, science, chart, detection, grounding, counting, and OCR. With the rule-based filtering and difficulty-based filtering, it yields 47.7K high-quality samples across 18 datasets and 8 tasks, including approximately 20.6K perception and 27.1K reasoning samples, primarily consisting of single-image, single-turn conversations. More specific information about the datasets are list in Tab 1.

**Implementation.**We train all models on the 18 datasets for 1 epochs using a learning rate of 1e-6, with a $5\%$ warmup before being held constant. We perform unified reinforcement training on Qwen2.5-VL-7B-instruct, comparing the GRPO and with our proposed variants, DualRPO. Since the vanilla GRPO uses a reference KL penalty which could be removed for better results(Yu et al. (2025)), we remove this component in the implementation of GRPO and DualRPO. We set $\mu = 0.01$ and $\lambda = 0.005$ to rescale the self-certainty signal such that it is brought to the same scale as the verifiable reward. We set $\mu$ higher than $\lambda$ to place greater emphasis on rewarding correct answers. For the KL loss, we set $\beta_{min} = 0.05$.

**Sample-level reward computation.** Traditional unified reinforcement training typically relies on step-wise optimization, which requires each batch to contain a single type of data. To enable a

more general training pipeline, we adopt the sample-level reward computation strategy proposed by Ma et al. (2025). This sample-level approach offers substantial flexibility and modularity, greatly simplifying the addition of new tasks or the modification of reward logic without altering the core training framework.

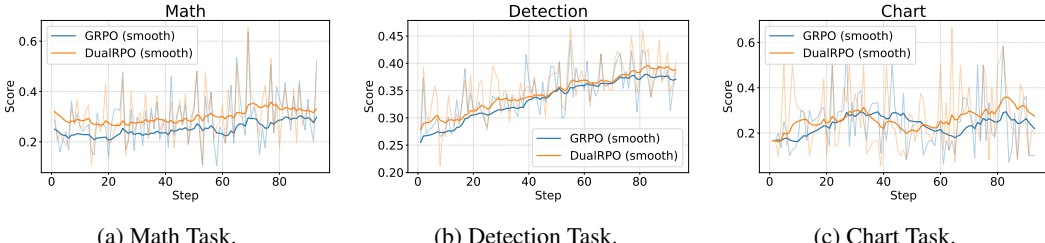

(a) Math Task.                    (b) Detection Task.                    (c) Chart Task.

Figure 3: **Comparison of the training dynamics upon the accuracy reward.** Solid lines indicate running averages with a smooth coefficient of 0.9. Notably, in all the three tasks, DualRPO demonstrates consistently faster learning from the early stages on GRPO.

**Evaluation.** To systematically evaluate the effectiveness of DualRPO, we conduct experiments on eight benchmarks that cover diverse multimodel perception and reasoning problems, including: MathVision(Wang et al. (2024)), MathVerse(Zhang et al. (2024)), MathVista(Lu et al. (2023)), MME(Fu et al. (2023)), development subset of $\text{MMBench}_{\text{EN}}$(Liu et al. (2024a)), SeedBench(Li et al. (2024b)), SeedBench2-Plus(Li et al. (2024b)), and OCRBench(Liu et al. (2024b)). The results are shown in Tab. 2, Tab. 3, and Tab. 4.

Table 1: We use 47.7K samples for our unified training. It includes two categories: reasoning and perception datasets, containing eight basic tasks with 18 datasets.

| type | task | dataset | size |
|---|---|---|---|
| reasoning | math | MM-Math(Sun et al. (2024)) | 3539 |
| | | Geometry3k(Lu et al. (2021)) | 2983 |
| | | MMK12 (Meng et al. (2025)) | 5732 |
| | Puzzle | PuzzleVQA (Chia et al. (2024)) | 2648 |
| | | VisualPuzzles (Song et al. (2025)) | 342 |
| | Science | ScienceQA(Lu et al. (2022)) | 536 |
| | | SciVQA (Borisova et al. (2025)) | 1264 |
| | | ViRL39K(science) (Wang et al. (2025a)) | 2539 |
| perception | Chart | ChartQAPro (Masry et al. (2025)) | 498 |
| | | ChartX (Xia et al. (2024)) | 2353 |
| | | Table-VQA (Kim et al. (2024)) | 493 |
| | | ViRL39K(chart) (Wang et al. (2025a)) | 1657 |
| | Detection | V3Det (Xie et al. (2023)) | 4000 |
| | | Object365 (Shao et al. (2019)) | 4000 |
| | Grounding | $D^3$ (Xie et al. (2023)) | 4870 |
| | Counting | CLEVR (Johnson et al. (2017)) | 1725 |
| | OCR | LLaVA-OV (Li et al. (2024a)) | 3092 |
| | | EST-VQA (Wang et al. (2020)) | 2946 |

Table 2: Performance of Qwen2.5-VL-7B-instruct trained with GRPO and DualRPO on different benchmark.

| | General | | Math | | | Perception |
|---|---|---|---|---|---|---|
| | MME | MMBench$_{EN}$ | MathVisoin$_{mini}$ | MathVista$_{mini}$ | MathVerse$_{mini}$ | OCRBench |
| Base | 2307 | 79.7 | 25.0 | 69.2 | 57.74 | 88.1 |
| +GRPO | 2330 | 80.4 | 26.6 | 69.8 | 58.34 | 88.4 |
| +DualRPO | 2344 | 80.2 | 28.3 | 70.8 | 58.90 | 88.5 |

Table 3: Performance of Qwen2.5-VL-7B-instruct trained with GRPO and DualRPO on SeedBench. This benchmark including several sub-task. For our unified model, we list the results about visual reasoning, text understanding, scene understanding, and counting for our comparison.

| SeedBench | Visual Reasoning | Text Understanding | Scene Understanding | Counting | Overall |
|---|---|---|---|---|---|
| Base | 84.6 | 73.8 | 80.0 | 71.1 | 76.9 |
| + GRPO | 83.1 | 72.8 | 79.5 | 71.0 | 76.2 |
| + DualRPO | 84.7 | 76.2 | 80.4 | 71.4 | 77.8 |

## 5 DISCUSSION

### 5.1 INTERNAL SIGNALS SERVE AS ONLY REWARD FOR UNIFIED TRAINING

Finding a universal, concise reward function for unified training remains the ultimate goal in this field. Yet, task heterogeneity typically demands multiple distinct reward functions to evaluate samples. Given the unique properties of internal signals, a natural question arises: can they act as the sole reward signal across all tasks? We investigated this via experiments and found this approach infeasible. While using the internal signal only for all tasks are elegant, sole reliance on them during training causes model collapse, this problem is particularly severe in tasks requiring strictly formatted outputs (e.g., detection) as shown in Fig.4a . We observed that using internal rewards as the sole signal leads to cross-task interference, which pushes outputs away from expected answers and ultimately derails training. For other tasks, sole reliance on internal rewards still yields suboptimal outcomes.

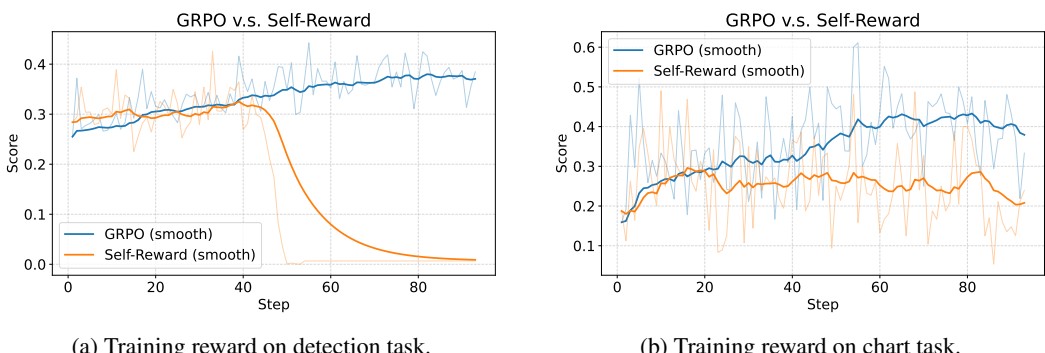

(a) Training reward on detection task.      (b) Training reward on chart task.

Figure 4: **Solo Internal rewards.** Using internal signals as the sole reward leads to different results: training on detection collapses, whereas training on the chart dataset is stable but attains rewards far below those from external supervision.

### 5.2 INTERNAL SIGNALS SERVE AS ONLY REWARD FOR PERCEPTION TASKS

While internal signals still cannot serve as the sole reward in unified training perfectly, we turn to exploring their effectiveness on subtasks (e.g., perception or reasoning) where task interference

Table 4: Performance of Qwen2.5-VL-7B-instruct trained with GRPO and DualRPO on SeedBench2-Plus. This benchmark including chart, map, and web understanding tasks.

| SeedBench2-Plus | Chart | Map | Web | Overall |
|---|---|---|---|---|
| Base | 67.53 | 60.22 | 83.48 | 69.56 |
| + GRPO | 67.16 | 60.01 | 82.71 | 69.11 |
| + DualRPO | 67.77 | 60.45 | 83.67 | 69.77 |

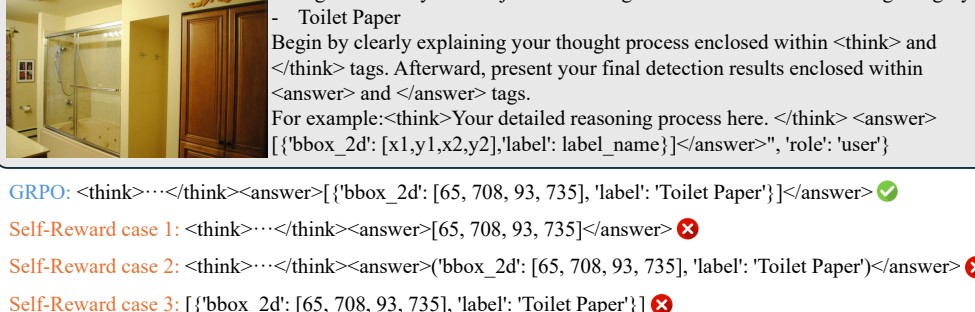

GRPO: <think>···</think><answer>[{'bbox_2d': [65, 708, 93, 735], 'label': 'Toilet Paper'}]</answer> ✅

Self-Reward case 1: <think>···</think><answer>[65, 708, 93, 735]</answer> ❌

Self-Reward case 2: <think>···</think><answer>('bbox_2d': [65, 708, 93, 735], 'label': 'Toilet Paper')</answer> ❌

Self-Reward case 3: [{'bbox_2d': [65, 708, 93, 735], 'label': 'Toilet Paper'}] ❌

Figure 5: Unlike models trained with GRPO, which maintain a consistent format due to the format reward, models rewarded by internal signals often miss vital formatting requirements.

is eliminated. To this end, we first examine training exclusively on perception data, with internal signals as the sole reward source. We find that with careful tuning, internal signals can act as a stable reward for the model—notably, the results even outperform those achieved with externally supervised rewards. We fine-tune Qwen2.5-VL-7B-Instruct on the VisionReasoner-7k dataset (Liu et al. (2025c)), which includes three task types: detection, segmentation, and counting. We evaluate on three benchmarks—RefCOCO, ReasonSeg, and Count—and report gIoU for segmentation tasks. The results demonstrate that training with internal rewards alone already outperforms the base model; surprisingly, it even exceeds external reward-based training on certain tasks.

Table 5: Results on perception tasks with different rewards. Learning from the internal reward manifest potential for the perception tasks.

| | RefCOCO$_{det}$ | RefCOCO$_{seg}$ | ReasonSeg | Count |
|---|---|---|---|---|
| Base | 87.54 | 75.62 | 56.65 | 76.33 |
| + GRPO | 88.70 | 75.90 | 63.41 | 81.31 |
| + Internal Reward | 88.88 | 76.47 | 61.34 | 82.21 |

## 5.3 INTERNAL SIGNALS SERVE AS ONLY REWARD FOR REASONING TASKS

The promising results trained on perception datasets reveal that the internal reward could serve as a powerful method to enhance the model's ability. We try to generalize the internal reward to the reasoning task. We collect three math datasets for our training: Geo3k, MMK12, and MM-Math as our training datasets, and use the same base model. However, the training results are not as good as the results in perception datasets.

## 5.4 INTERNAL SIGNALS SUFFICE AS THE ONLY REWARD FOR REASONING TASKS

Internal signals as the sole reward yield distinct results for reasoning and perception tasks, and analyzing policy entropy provides a complementary explanation. For perception tasks, policy entropy is high—signaling low confidence in the model's outputs. Using an internal, self-certainty-based internal reward encourages more confident, consistent predictions, delivering clear performance gains.

Table 6: Results on reasoning tasks with different rewards. Although internal signals could serve a possible reward for perception tasks, it is hard to use the same reward for the reasoning tasks.

|                  | MathVision$_{\text{mini}}$ | MathVista$_{\text{mini}}$ | MathVerse$_{\text{mini}}$ |
| ---------------- | -------------------------- | ------------------------- | ------------------------- |
| Base             | 25.0                       | 69.2                      | 57.7                      |
| + GRPO           | 28.6                       | 76.3                      | 66.4                      |
| + Internal Reward| 24.8                       | 70.3                      | 58.3                      |

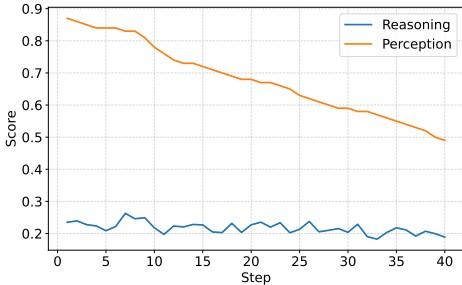

Figure 6: **Entropy loss comparison of perception and reasoning datasets.** We find the when training on perception datasets with higher initial entropy loss could be easily optimized with only the internal reward. One the other hand, instruction-tuning model which has lower entropy on reasoning tasks can not be optimized with it.

By contrast, the instruction-tuned base model begins with relatively low entropy for reasoning tasks—it already generates confident answers. Consequently, the self-certainty signal provides only limited additional reward shaping, and external rewards offer minimal value beyond what the model already leverages. This discrepancy in initial entropy helps explain why internal signals alone are insufficient for reasoning tasks.

## 6 CONCLUSION

We presented DualRPO, a reinforcement learning framework for vision–language models that combines external verifiable rewards with internal confidence signals. By leveraging self-certainty, the divergence between the model's predictions and a uniform distribution, we modulate both the reward function and the KL penalty, reducing training instability and alleviating task interference across heterogeneous domains. Our study highlights the dual role of internal signals. When used alone under unified training, they fail to prevent collapse, especially for format-sensitive perception tasks. Yet in isolated perception settings, internal rewards are stable and effective, even surpassing external verifiers. Reasoning tasks, however, benefit less from self-certainty alone, as lower entropy limits its impact. This contrast suggests that internal signals are valuable but not sufficient universally, providing complementary rather than standalone benefits.

Through extensive experiments on eight reasoning and perception sub-tasks with Qwen2.5-VL-7B-Instruct, we demonstrate that DualRPO consistently outperforms GRPO baselines. Certainty-guided modulation amplifies correct but uncertain responses, penalizes overconfident errors, and adaptively adjusts KL regularization. These mechanisms improve stability, sharpen predictions, and enhance cross-task generalization.

In summary, DualRPO makes three contributions: (1) introducing confidence-aware modulation that unifies internal and external signals for reinforcement learning in VLMs; (2) providing a systematic study of when internal rewards succeed or fail; and (3) delivering stable performance improvements across diverse tasks. We believe confidence-guided reward modulation offers a promising principle for advancing robust and generalizable reinforcement learning in multimodal systems.

**Declaration of LLM Usage** During the writing of this thesis, authors primarily used a large language model (LLM) to assist with text polishing and grammar correction. All research ideas, academic arguments, and conclusions were independently developed by the authors.

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
