# OpenReview forum: "DualRPO: All-in-one Visual RL with Internal and External Rewards"
_ICLR.cc/2026/Conference — Submitted to ICLR 2026_

### Official Review · Reviewer_EJmr · 2025-10-29

**Soundness:** 2
**Presentation:** 2
**Contribution:** 2
**Rating:** 4
**Confidence:** 4

**Summary:**

This paper proposes DualRPO, a reinforcement learning framework that unifies perception and reasoning tasks in VLM. DualRPO combines external verifiable rewards with internal self-certainty rewards, defined as the KL divergence between the model’s output distribution and a uniform distribution. The method amplifies rewards for correct but low-confidence outputs, penalizes overconfident errors, and dynamically adjusts the KL regularization term based on confidence.

**Strengths:**

**Clear motivation**: The paper clearly identifies the key challenge of multi-task RL training for VLMs and directly targets this issue by designing a framework to enable stable joint optimization across perception and reasoning tasks.

**Intuitive idea**: The proposed DualRPO framework is conceptually intuitive—by integrating internal self-certainty rewards with external verifiable rewards into a unified optimization process, it naturally aligns the model’s confidence with correctness, achieving a balanced improvement in both training stability and task performance.

**Clear structure**: The paper is well organized, presenting the motivation, method, and experiments in a logical order. The description of the proposed approach is straightforward and easy to follow.

**Weaknesses:**

**Limited effectiveness**: The improvements reported in Tables 2 and 3 are minor, with only small gains over the baseline and several results even worse than GRPO. These marginal differences do not clearly demonstrate the effectiveness of the proposed DualRPO method.

**Missing perception-task results**: Although the paper claims that DualRPO is effective for unified multi-task RL and the training set includes many perception-related datasets (as shown in Table 1), there are no reported results for DualRPO on perception tasks. The only related experiment is Section 5.2 on internal signals as the sole reward, which does not validate the full DualRPO framework. Without such results, the claim of DualRPO’s general applicability to perception tasks remains unsubstantiated.

**Limited novelty**: The idea of using internal signals or confidence-based rewards has been explored in several prior works. The contribution here mainly lies in combining these ideas rather than introducing a fundamentally new mechanism.

**Unfocused discussion**: The Discussion section devotes extensive space to analyzing internal signals as the only reward, which diverges from the paper’s main theme of dual rewards. This part feels redundant and does not effectively highlight the advantages or distinct contributions of the DualRPO framework.

**Questions:**

1.Please provide experimental results of DualRPO on perception tasks. This is crucial to verify whether the dual-reward design truly benefits multi-task RL that mixes reasoning and perception domains.

2.The KL modulation based on self-certainty is an important part of your method. Could you show an ablation study that isolates the effect of reward modulation and KL modulation separately?

3.In Tables 2 and 3, the performance differences from GRPO are relatively small. Could you provide variance or statistical significance analysis to confirm these improvements are consistent across runs?

4.The paper emphasizes confidence calibration. Have you evaluated this explicitly to support the claim that DualRPO improves model confidence?

5.Other questions refer to weakness

---

### Official Review · Reviewer_TdPe · 2025-10-29

**Soundness:** 3
**Presentation:** 2
**Contribution:** 3
**Rating:** 4
**Confidence:** 4

**Summary:**

This paper introduces DualRPO, a novel reinforcement learning paradigm designed to address task interference in unified multi-task training for Vision-Language Models (VLMs). By synergistically integrating internal self-certainty rewards and external verifiable rewards, DualRPO demonstrates significant performance improvements across diverse perceptual and cognitive tasks. The research provides valuable insights into the role of internal signals in multimodal RL.

**Strengths:**

1. DualRPO integrates model self-certainty as an internal signal with external verifiable rewards for reinforcement learning in VLMs. This dual-modulation design (for both reward shaping and KL loss) offers a promising avenue for mitigating task interference in multi-task training and successfully achieves performance enhancements.

2.The paper validates DualRPO's effectiveness through extensive experiments across  heterogeneous tasks (encompassing both perceptual and cognitive types). DualRPO consistently achieves significant performance gains compared to the GRPO baseline,

**Weaknesses:**

1.The paper identifies task interference as a "critical bottleneck" in unified VLM training and states that DualRPO aims to "mitigate" it.  However, a deeper definition, quantification, and explanation of how DualRPO's mechanisms specifically address the underlying causes of task interference are lacking: The paper does not provide a clear operational definition or specific metrics to quantify the degree of task interference.
2.Although the paper claims that DualRPO substantially reduces the amplitude of training instability and presents smoothed training curves (e.g., Figure 3), a visual inspection of the provided Step-Score graphs suggests that the step-to-step score fluctuations for DualRPO are not significantly smaller than those of GRPO. and  DualRPO introduces crucial hyperparameters . The paper specifies the values used but does not provide an analysis of their sensitivity to model performance and training stability. This omission limits the verification of the method's generalizability and robustness.
3.The paper exhibits inconsistencies in referencing figures and tables, which affects its professional presentation and readability. For instance, some figures (e.g., Figure 6)   appear without an explicit in-text reference statement guiding the reader to them.

**Questions:**

Please refer to the Weaknesses section above for details.

---

### Official Review · Reviewer_Ddfm · 2025-10-31

**Soundness:** 2
**Presentation:** 2
**Contribution:** 1
**Rating:** 2
**Confidence:** 4

**Summary:**

DualRPO introduces a GRPO-style VLM training scheme that combines an internal self-certainty reward, defined as the average KL divergence between next-token probs and a uniform distribution, with external verifiable rewards. The self-certainty scales both (i) the task reward (amplify correct/uncertain; penalize wrong/overconfident) and (ii) the GRPO KL term (weaker when confident, stronger when uncertain), aiming to stabilize unified multi-task VLM training across perception and reasoning tasks.

**Strengths:**

- The paper precisely specifies the internal reward (token-wise KL to uniform) and how it modulates rewards and KL in GRPO; the optimization objective is explicit.
- The paper finds that internal-only reward can collapse for format-sensitive perception tasks; it suits better on some perception subtasks, and worse on reasoning, clarifying when internal signals suffice.

**Weaknesses:**

-  Incremental novelty relative to LM literature: The self-certainty/entropy idea is established for language models, e.g., INTUITOR [1]; this paper mainly combines it for VLMs via linear reward/KL scalings. Besides, internal/intrinsic rewards are long-standing in RL [2][3].
- Missing reward-only vs. KL-only vs both ablations; with/without reference-KL vs the proposed KL schedule. This makes it hard to attribute gains to the proposed dual modulation rather than tuning choices.
- The abstract claims reduced “training instability amplitude” but provides neither a definition nor a number; main text lacks a clear, quantitative stability metric with statistics.

[1] Learning to Reason without External Rewards, arXiv:2505.19590

[2] Curiosity-driven Exploration by Self-supervised Prediction arXiv:1705.05363

[3] Self-Supervised Exploration via Disagreement arXiv:1906.04161

**Questions:**

1. Please report reward-only, KL-only, and both; and sensitivity to hyperparameters. Also compare against GRPO with standard reference-KL to show DualRPO’s KL schedule is beneficial beyond removing reference-KL.
2. Define the “training instability amplitude” precisely and quantify its reduction.
3. It would be better if ≥3 seeds experiments can be provided with means/std for all main tables; highlight which gains are statistically significant.
4. How does DualRPO's internal reward fundamentally differ from INTUITOR beyond the input modality?

---

### Meta-Review · Area_Chair_FWRK · 2026-01-07

**Summary:**

The paper received uniformly negative reviews, with no discussion or author response available. Based on the reviewers’ feedback, the area chair recommends rejection.

**Reviewer Scores:**

n/a

---

### Decision · Program_Chairs · 2026-01-26

Reject